# Case study of Argus in Togo: An SMS and web-based application to support public health surveillance, results from 2016 to 2019

José Guerra[1]*, Kokou Mawule Davi[2], Florentina Chipuila Rafael[1], Hamadi Assane[3], Lucile Imboua[2], Fatoumata Binta Tidiane Diallo[2], Tsidi Agbeko Tamekloe[4], Aklagba Kuawo Kuassi[2], Farihétou Ouro-kavalah[3], Ganiou Tchaniley[5], Nassirou Ouro-Nile[5], Pierre Nabeth[1]

1 Health Emergencies Programme, World Health Organization, Lyon, France, 2 Country Office, World Health Organization, Lomé, Togo, 3 Division de la Surveillance Intégrée des Urgences Sanitaires et Riposte, Ministère de la Santé, Lomé, Togo, 4 Direction de la lutte contre la maladie et des Programmes de santé publique, Ministère de la Santé, Lomé, Togo, 5 Direction du système national d'information sanitaire, Ministère de la Santé, Lomé, Togo

* j.guerra.sp@gmail.com

## Abstract

### Introduction

Argus is an open source electronic solution to facilitate the reporting and management of public health surveillance data. Its components include an Android-phone application, used by healthcare facilities to report results via SMS; and a central server located at the Ministry of Health, displaying aggregated results on a web platform for intermediate and central levels. This study describes the results of the use of Argus in two regions of Togo.

### Methods

Argus was used in 148 healthcare facilities from May 2016 to July 2018, expanding to 185 healthcare facilities from July 2018. Data from week 21 of 2016 to week 12 of 2019 was extracted from the Argus database and analysed. An assessment mission took place in August 2016 to collect users' satisfaction, to estimate the concordance of the received data with the collected data, and to estimate the time required to report data with Argus.

### Results

Overall completeness of data reporting was 76%, with 80% of reports from a given week being received before Tuesday 9PM. Concordance of data received from Argus and standard paper forms was 99.7%. Median time needed to send a report using Argus was 4 minutes. Overall completeness of data review at district, regional, and central levels were 89%, 68%, and 35% respectively. Implementation cost of Argus was 23 760 USD for 148 facilities.

WHO logo is not permitted. This notice should be preserved along with the article's original URL.

**Data Availability Statement:** All files are available in a supporting information file.

**Funding:** The pilot-test of Argus in Togo and its evaluation were jointly funded by grants from the French Republic and the Russian Federation. The funders had no role in study design, data collection and analysis, decision to publish, or preparation of the manuscript.

**Competing interests:** The authors have declared that no competing interests exist.

## Conclusions

The use of Argus in Togo enabled healthcare facilities to send weekly reports and alerts through SMS in a user-friendly, reliable and timely manner. Reengagement of surveillance officers at all levels, especially at the central level, enabled a dramatic increase in completeness and timeliness of data report and data review.

## Introduction

Public health surveillance is an essential function of a health system, defined as "the systematic on-going collection, collation and analysis of data for public health purposes and the timely dissemination of public health information for assessment and public health response as necessary" [1]. Since 1998, the World Health Organization (WHO) has advocated for the Integrated Disease Surveillance and Response (IDSR) approach for public health surveillance in the WHO African region [2, 3] with specific roles defined at each level of the surveillance system. In resource-limited settings, paper-based transmission is the traditional way of reporting weekly public health surveillance data [3, 4]. Due to lack of resources and infrastructure, paper-based transmission may hamper full and timely data reporting and lead to difficulties in the routine analysis of the collected data, ultimately delaying the detection of a potential public health event such as an outbreak [3–8].

In the last few decades, the development of telecommunication has created an opportunity to enhance both the completeness and timeliness of data reporting. While information technology (IT) can facilitate the management and analysis of collected data [9], striking development of mobile phone infrastructure in resource-limited settings, including in hard-to-reach areas, along with internet access at the intermediate and central levels of the surveillance system offer new possibilities for IT tools to facilitate public health surveillance in compliance with the IDSR. Dedicated IT tools have already been used in several countries to facilitate reporting and management of public health surveillance data [7, 9–13]. In practice, many systems failed to allow remote healthcare facilities to report data due to lack of internet connectivity. Furthermore, we were unable to identify a single tool which would allow each level of the system to review and analyse data from its area in a workflow respecting the IDSR procedures.

WHO has therefore developed "Argus", an IT tool to facilitate public health surveillance for early detection and response of events in compliance with IDSR procedures (http://www.argus.community/). It uses Short Message Service (SMS) technology for the transmission of aggregated weekly reports and alerts of serious or unexpected events between healthcare facilities and all levels of the public health surveillance system via a mobile phone application. A web platform complements the phone application for data management and analysis at each level of the system.

To finalize the development of Argus and assess if it could improve public health surveillance in Togo, Argus had been pilot tested in two regions of the country since May 2016. In this paper, we describe the results of the Argus pilot test in Togo in terms of data reporting by healthcare facilities, data review by surveillance officers, public health events identified, users' satisfaction and costs.

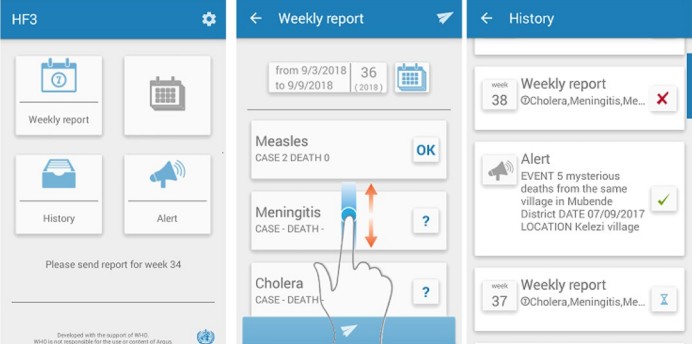

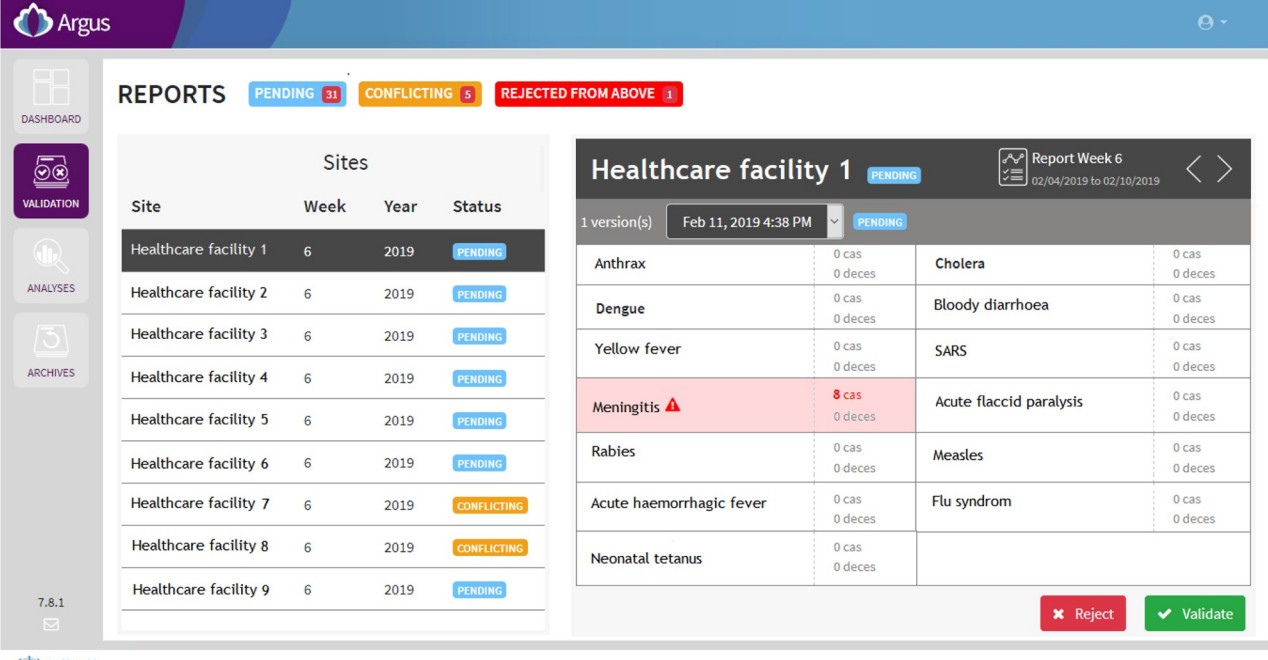

**Fig 1. Argus IT solution for public health surveillance.**

## Material and methods

### Argus

Argus included several components (http://www.argus.community/):

- A custom Argus Android Client application installed on Android mobile phones, Fig 1.

- An Argus server located at the Togolese Ministry of Health in charge of data collection and management. It displayed an online web platform to review the collected data and identify potential acute public health events through weekly epidemiological summaries and dedicated dashboards, Fig 1. The Argus server was composed of three PHP applications exchanging information through XML files and web services, a web front end developed with the AngularJS framework, and an instance of the R statistical software.

- A custom Argus Android Gateway application was installed on eight Android mobile phones at the Togolese Ministry of Health to act as a gateway between the Argus Android Clients and the Argus server.

Argus was released with a GNU AGPL3 license.

## Pilot test in Togo

Argus was implemented in two regions of Togo (Lomé Commune and Savanes) in May 2016 to assess its usefulness in improving public health surveillance. Lomé Commune is the urban region capital of the country, located in the south. The other region, Savanes, is the northernmost region of the country, consisting mainly of rural and hard-to-reach areas.

Before implementing Argus, weekly public health surveillance from healthcare facilities was performed via paper transmission or phone calls to the district level. The district-aggregated forms were transmitted to the regions, who in-turn transmitted region-aggregated forms to the central level.

To make the best use of limited human and financial resources for public health surveillance, the list of events to be weekly reported should be kept to a minimum to ensure that it is manageable by the system and that adequate resources are available to carry out a response. For early detection of outbreaks via weekly aggregated reporting, it was recommended to keep the number of variables to a minimum, ideally reporting only number of cases and deaths, to avoid unnecessary burden on the healthcare facilities and maximize reporting efficiency. In the planning process of Argus, and prior to its implementation, the Ministry of Health revised the list of diseases and conditions to be reported on a weekly basis. It was reduced from 20 to 13, with acute flaccid paralysis, acute haemorrhagic fever, bloody diarrhoea, cholera, flu syndrome, human anthrax, human rabies, measles, meningitis, neonatal tetanus, other severe diarrhoea, severe acute respiratory syndrome and yellow fever. The variables to be collected were reduced from seventeen variables (mainly number of cases and deaths by age groups) to two: the weekly number of cases and weekly number of deaths. A new paper form was developed to support tallying of the weekly figures based on the healthcare facility registry.

Data was to be reported with Argus before the paper form was archived in the healthcare facility. Alert thresholds at healthcare facility level were defined for nine diseases and conditions (acute flaccid paralysis: 1, acute haemorrhagic fever: 1, cholera: 1, human anthrax: 1, human rabies: 1, measles: 3, meningitis: 2, neonatal tetanus: 1, yellow fever: 1). If the weekly number of cases reached an alert threshold, a warning message appeared on the Argus Android Client application at the healthcare facility level and the disease or condition was highlighted on the aggregated report displayed on the Argus web platform.

Argus duties at each level of the surveillance system were defined as follows, based upon the IDSR guidelines:

- Healthcare facility: send the weekly aggregated report from the previous week on Monday before four PM. In case of occurrence of a serious or unexpected event, staff at healthcare facilities were required to immediately notify their supervisor, they could do so using Argus. Upon description of the serious or unexpected event on the Argus Android Client application, information was immediately forwarded to the personal mobile phone of pre-identified supervisors by SMS.

- District level: review (i.e. validate or reject) healthcare facilities weekly reports and examine the weekly epidemiological summary to identify any potential public health event before Monday midnight.

- Regional level: review districts aggregated weekly reports and examine the weekly epidemiological summary before Tuesday four PM.

- Central level: review regions aggregated weekly reports and examine the weekly epidemiological summary before Tuesday midnight.

Seven training sessions were conducted between the 19[th] and 25[th] May 2016, attended by at least one staff from each healthcare facility, one surveillance officer from each district and region, and one surveillance officer from the central level. An Android phone equipped with Argus and a parental control application was provided to each healthcare facility along with brochures and posters detailing how to use it. Due to the parental control application, the phone could only be used to run the Argus application. Each training session was composed of:

- A 1-hour plenary on the concepts and objectives of public health surveillance;

- A 30-minute plenary on the Argus tool and its pilot in Togo;

- A 45-minute practical exercise on sending a weekly report and an alert; and

- A 1-hour session for intermediate and central level surveillance officers on the use of the central web platform to review reports, examine their weekly epidemiological summary and perform custom analyses.

Deployment of Argus took place in two phases. An initial phase was conducted from May 2016 to July 2018 and included 148 healthcare facilities (54 from the 5 districts of Lomé Commune and 94 from the 5 districts of Savanes). A reinforcement phase then started on July 2018, including 185 healthcare facilities (87 from Lomé Commune and 98 from Savanes). The implementation, operation and assessment of the pilot were conducted under the supervision of the Togolese Ministry of Health with the support of the WHO country office in Togo and WHO Headquarters Lyon office.

Prior to the reinforcement phase, the list of diseases and conditions to be reported and the alert thresholds were updated by the Ministry of Health. During this update, other severe diarrhoea was removed; dengue, neonatal deaths and maternal deaths were added; alert threshold for measles was reduced to 2 cases, and alert thresholds for bloody diarrhoea and flu syndrome were added with 5 and 20 cases respectively. Seven training sessions were conducted in July 2018 to update the Argus application on all healthcare facility phones and provide reinforcement training to all staff. A responsible focal point at the central level was designated to supervise the pilot of Argus during the reinforcement phase.

## Outcome measures

An evaluation protocol detailing the indicators to be collected and their collection and analysis modalities was developed by the WHO and agreed upon with the Ministry of Health at the onset of the pilot test. The primary outcomes of interest were the fulfilment of the duties attributed to each level of the surveillance system, i.e. the completeness and timeliness of data reporting and data review. Additional indicators were added to assess: the quality of the reported data (i.e. concordance of the received data with the collected data); the public health events identified (i.e. number of alert thresholds reached and immediate notifications received); the users' satisfaction regarding several characteristics of the system (i.e. appearance, documentation, usefulness, simplicity); the reliability of the system (i.e. percentage of uptime of the Argus web platform); and the costs of implementation and operation. The full list of

**Table 1. Indicators used to describe the Argus pilot test results.**

| Type of outcome measure | Indicators and calculation method | Data source | Period |
|---|---|---|---|
| Data reporting by healthcare facilities | Evolution of completeness and timeliness of data reporting by healthcare facilities. | Argus database. | W21 2016 –W12 2019 |
| | *(No. of reports received from healthcare facilities / No. of reports expected[a] from healthcare facilities)*100.* | | |
| | *(No. of reports received on time from healthcare facilities/ No. of reports expected[a] from healthcare facilities)*100.* | | |
| | Time required by a user at healthcare facilities to report data with Argus Android client. | Assessment mission. | |
| | Concordance of the received data with the collected data. | Paper forms. | W21 2016 –W31 2016 |
| | *(No. of data having the same value in the system and on the paper forms) / (No. of data collected in the system)*100.* | Argus database. | |
| Data review by surveillance officers | % of uptime of the Argus web platform. | Monitoring service[b]. | 30/05/2016–01/04/2019 |
| | *(duration of Argus web platform online and available to users / duration of Argus web platform offline)*100.* | | |
| | Evolution of completeness and timeliness of data review at each level. | Argus database. | W21 2016 –W12 2019 |
| | *By level: (No. of validated or rejected reports / No. of reports received)*100.* | | |
| | *(No. of reports received and validated or rejected on time / No. of reports received on time)*100.* | | |
| Public health events identified | Number of reached alert thresholds identified by the system. | Argus database. | W21 2016 –W12 2019 |
| | Number of immediate notifications of serious or unexpected events received. | Argus database. | 30/05/2016–01/04/2019 |
| Users' satisfaction | Opinion on: | Assessment mission. | |
| | • general appearance of Argus Android Client at healthcare facilities; | | |
| | • available documentation for Argus Android Client used at healthcare facilities; | | |
| | • general appearance of Argus web platform at intermediate and central levels; | | |
| | • available documentation for Argus web platform at intermediate and central levels; | | |
| | • usefulness of Argus Android Client used for data reporting; | | |
| | • usefulness of Argus web platform for data review; | | |
| | • usefulness of Argus web platform for data analysis; | | |
| | • overall simplicity of use of Argus Android client; | | |
| | • simplicity of use of Argus Android client for data reporting; | | |
| | • overall simplicity of use of Argus web platform; | | |
| | • simplicity of use of Argus web platform for data review; | | |
| | • simplicity of use of Argus web platform to monitor the completeness and timeliness of data reporting. | | |
| | *Sum of ratings assigned by respondents (from 1 worst to 5 best) / No. of respondents.* | | |
| Costs | Cost of setting up Argus. | Invoices paid for the project. | 18/05/2016–04/04/2019 |
| | Cost of internet access to the central server. | Argus database | |
| | Cost of SMS exchanges. | | |

[a] One report expected each week from each healthcare facility included in the pilot.

[b] https://www.site24x7.com/

indicators used to assess and describe the Argus pilot test results in Togo is presented in Table 1.

To assess data reporting and data review and to describe public health events identified by the system we extracted data from epidemiological week 21 of 2016 to epidemiological week 12 of 2019 from the Argus database. We monitored the percentage of uptime of the Argus web platform through an online monitoring service (https://www.site24x7.com). To describe the

costs of the implementation and operation of the system we used the invoices paid for the project and extracted the number of SMS exchanged from the Argus database.

One assessment mission took place from the 4th to 12th August 2016 to collect users' satisfaction, to estimate the concordance of the received data with the collected data, and to estimate the time required to report data with Argus. This was done through eight sessions organized with all users of the system, each session was composed of: a 20-minute presentation on the current use of Argus in Togo; a 40-minute slot for participants to fill in a user-satisfaction questionnaire (S1 Questionnaires); a 1-hour structured discussion with all participants; and a 30-minute structured discussion with surveillance officers at intermediate and central levels.

To estimate the concordance of the received data with the collected data, participants were asked to bring the paper forms archived at the healthcare facilities and used to tally the weekly figures. A copy of each form was collected and its data entered twice into a database using the Epidata software.

To estimate the time required to report data with Argus, participants in four of the eight evaluation sessions were provided with a mock paper tallying form and asked to report it using Argus. The amount of time it took each participant to enter the information and send the report was measured and registered by one surveyor.

All data analyses were performed using the R statistical software [14].

## Results

### Data reporting by healthcare facilities

Results of completeness and timeliness of data reporting by healthcare facilities are available Table 2.

The evolution of completeness of data reporting is presented in Fig 2, the distribution of weekly reports reception in Fig 3.

By district, for the overall period, completeness of data reporting ranged from 63% to 90% and timeliness from 36% to 59%.

The median time required by a user at a healthcare facility to fill and send a weekly report (composed of the number of cases and deaths for 13 diseases and conditions) was 4 minutes (n = 73; 10th- 90th percentiles: [2 minutes; 7 minutes]).

One thousand seventy-one paper forms were collected from 144 healthcare facilities. When comparing reports received from Argus with the paper forms: 94% were identical (n = 1200); 4% had one error (n = 48); 2% had more than one error (n = 23). The overall percentage of consistent data between Argus and the paper forms was 99.7% (n = 32397 / 32506).

### Data review by surveillance officers

From the 30th May 2016 to 1st April 2019, the Argus web platform was online and accessible to the users 94% of the time. The Argus web platform faced 61 offline periods: 11 lasted less than 10 minutes; 26 lasted between 10 minutes and one hour; 19 lasted between one hour and one day; five lasted more than a day (i.e. 2 days, 5 days, 15 days, 17 days, 20 days).

Results of completeness and timeliness of data review by surveillance officers are available Table 2. The evolution of completeness of data review is presented in Fig 4.

By district, for the overall period, the completeness and timeliness of data review ranged from 82% to 96% and 22% to 68% for the completeness and timeliness respectively.

**Table 2. Results of completeness and timeliness of data reporting and data review.**

| Indicator | Overall period | Initial phase | Reinforcement phase |
|---|---|---|---|
| Level | (weeks 21 of 2016–12 of 2019) | (weeks 21 of 2016–29 of 2018) | (weeks 30 of 2018–12 of 2019) |
| **Completeness of data reporting** | | | |
| *% (No. of reports received; No. of expected reports)* | | | |
| **Country** | 76% (17 406; 23 062) | 69% (11 591; 16 724) | 92% (5 815; 6 338) |
| **Savanes region** | 81% (11 315; 13 979) | 78% (8 240; 10 622) | 92% (3 075; 3 357) |
| **Lomé Communes region** | 67% (6 091; 9 083) | 55% (3 351; 6 102) | 92% (2 740; 2 981) |
| **Timeliness of data reporting** | | | |
| *% (No. of reports received before Monday 4PM; No. of expected reports)* | | | |
| **Country** | 48% (10 987; 23 062) | 39% (6 573; 16 724) | 70% (4 414; 6 338) |
| **Savanes region** | 52% (7 267, 13 979) | 46% (4 846; 10 622) | 72% (2 421; 3 357) |
| **Lomé Communes region** | 41% (3 720, 9 083) | 28% (1 727; 6 102) | 67% (1 993; 2 981) |
| **Median time for a weekly report to be received by the server** | | | |
| *Time from the end of the reported week, Sunday midnight (No. of received reports; [10th-90th percentiles])* | | | |
| | 11h (n = 17 221; [7h–10d]) | 13h (n = 11 406; [8h–14d]) | 9h (n = 5 815; [7h–7d]) |
| **Time when 80% of reports were received by the server**. | | | |
| *Time from the end of the reported week, Sunday midnight* | | | |
| | 2 d 21h (Wednesday 9PM) | 3d 19h (Thursday 7PM) | 1d 8h (Tuesday 8AM) |
| **Completeness of data review** | | | |
| **District level** | 89% (15 511; 17 406) | 86% (9 442; 11 591) | 96% (5 569; 5 815) |
| *% (No. of reports validated or rejected; No. of reports received from healthcare facilities)* | | | |
| **Regional level** | 68% (958; 1 404) | 60% (643; 1 065) | 93% (315; 339) |
| *% (No. of reports validated or rejected; No. of district aggregated reports)* | | | |
| **Central level** | 35% (101; 289) | 16% (37; 225) | 100% (64; 64) |
| *% (No. of reports validated or rejected; No. of regional aggregated reports)* | | | |
| **Timeliness of data review** | | | |
| **District level** | 48% (5 279; 10 987) | 31% (2 050; 6 573) | 73% (3 229; 4 414) |
| *% (No. of reports reviewed before Monday midnight; No. of reports received from healthcare facilities before Monday 4PM)* | | | |
| **Regional level** | 41% (188; 459) | 21% (46, 223) | 60% (142; 236) |
| *% (No. of reports reviewed before Tuesday 4PM; No. of district aggregated reports created before Monday midnight)* | | | |
| **Central level** | 43% (29; 67) | 7% (2; 27) | 68% (27; 40) |
| *% (No. of reports reviewed before Tuesday midnight; No. of regional aggregated reports created before Tuesday 4PM)* | | | |
| **Median time for a district to review a report upon reception (hours, [10th-90th percentiles])** | | | |
| *Time upon reception of the report; [10th-90th percentiles]* | | | |
| | 1d 7h (n = 16 135; [3h–19d]) | 2d 5h (n = 10 511; [6h–25d]) | 8h (n = 5 624; [1h–7d]) |

## Public health events identified

From week 21 of 2016 to week 12 of 2019, alert thresholds were reached 934 times: flu syndrome (n = 321); yellow fever (n = 251); meningitis (n = 88); acute flaccid paralysis (n = 71); bloody diarrhoea (n = 69); human rabies (n = 50); acute haemorrhagic fever (n = 38); measles (n = 36); neonatal tetanus (n = 9); cholera (n = 1). Ninety-six healthcare facilities reported a weekly number of cases reaching an alert threshold at least once. Sixty-four immediate notification of serious or unexpected public health events were sent by a healthcare facility and forwarded to the personal mobile phone of their supervisor. Eighteen were related to a real alert (i.e. four for suspect cholera, three meningitis, two acute haemorrhagic fever, one acute flaccid paralysis, one avian flu, one bloody diarrhoea, one cluster of deaths, one food poisoning, one

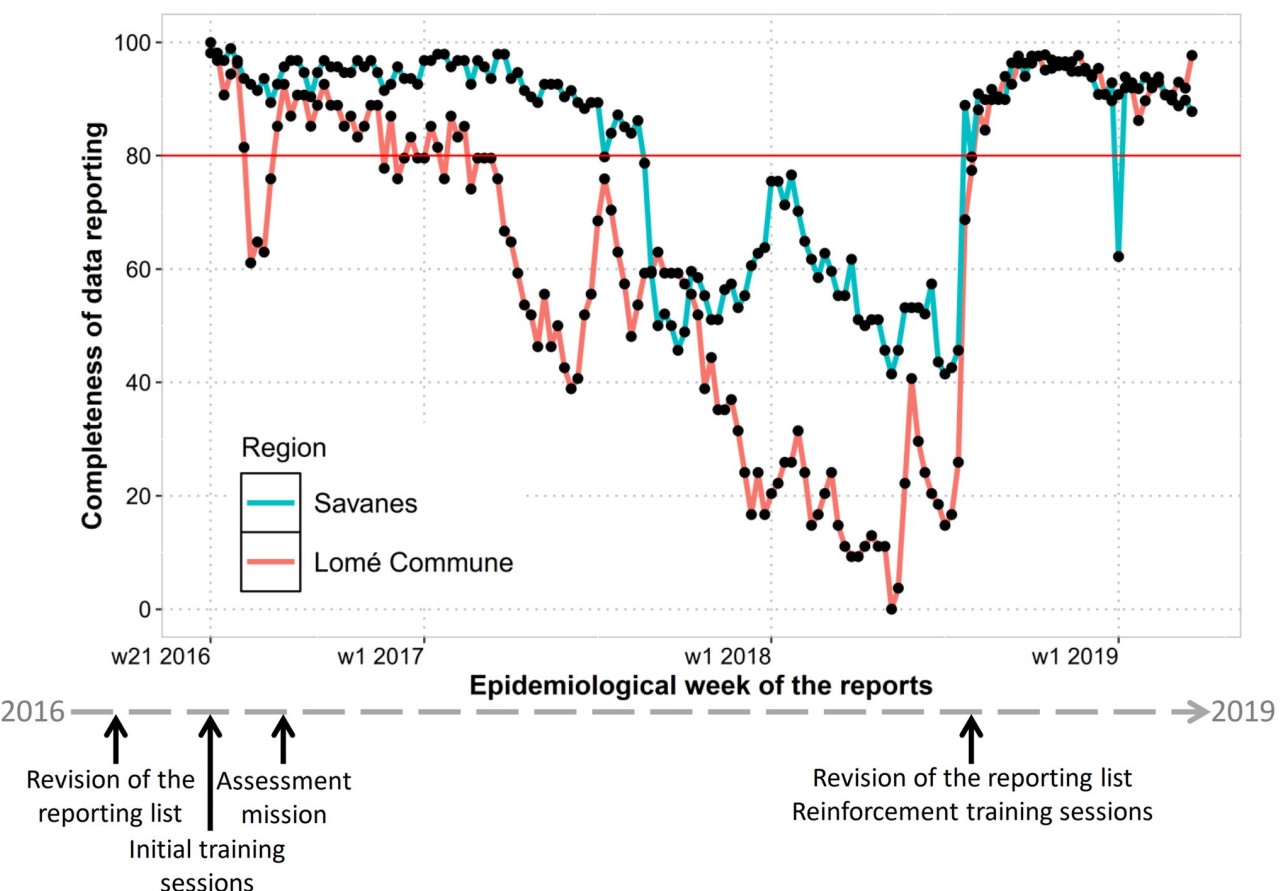

**Fig 2. Evolution of completeness of data reporting by healthcare facilities from week 21 2016 to week 12 2019 (n = 23 062 expected reports).**

human anthrax, one human rabies, one measles, one yellow fever) and 46 were related to testing or manipulation error by the healthcare facility.

## Users' satisfaction

An evaluation questionnaire was filled by 144 healthcare staff and 20 surveillance officers at district, regional and central levels. Ninety nine percent of the healthcare facility staff reported owing a personal mobile phone (n = 140/142), with a median duration of ownership of 10 years (10th-90th percentiles: [7 years; 15 years]). Ninety percent of the surveillance officers reported owing a personal computer (n = 18/20), with a median duration of ownership of 4 years (10th-90th percentiles: [3 years; 14 years]). All surveillance officers reported accessing the Internet for their personal use: 45% daily (n = 9/20), 35% weekly (n = 7/20), 20% less than weekly (n = 4/20). Results of the questionnaire are displayed in Table 3.

## Costs

The sum of implementation costs was 23 760 USD, shared between: two servers (4 351 USD), 185 Android phones (10 770 USD), SIM cards (655 USD), brochures and posters printing (855 USD), training sessions (7 130 USD).

The internet access of the central server (for the Argus web platform to be online) costed 168 USD per month. From 18th May 2016 to 04th April 2019, 732 932 SMS were exchanged

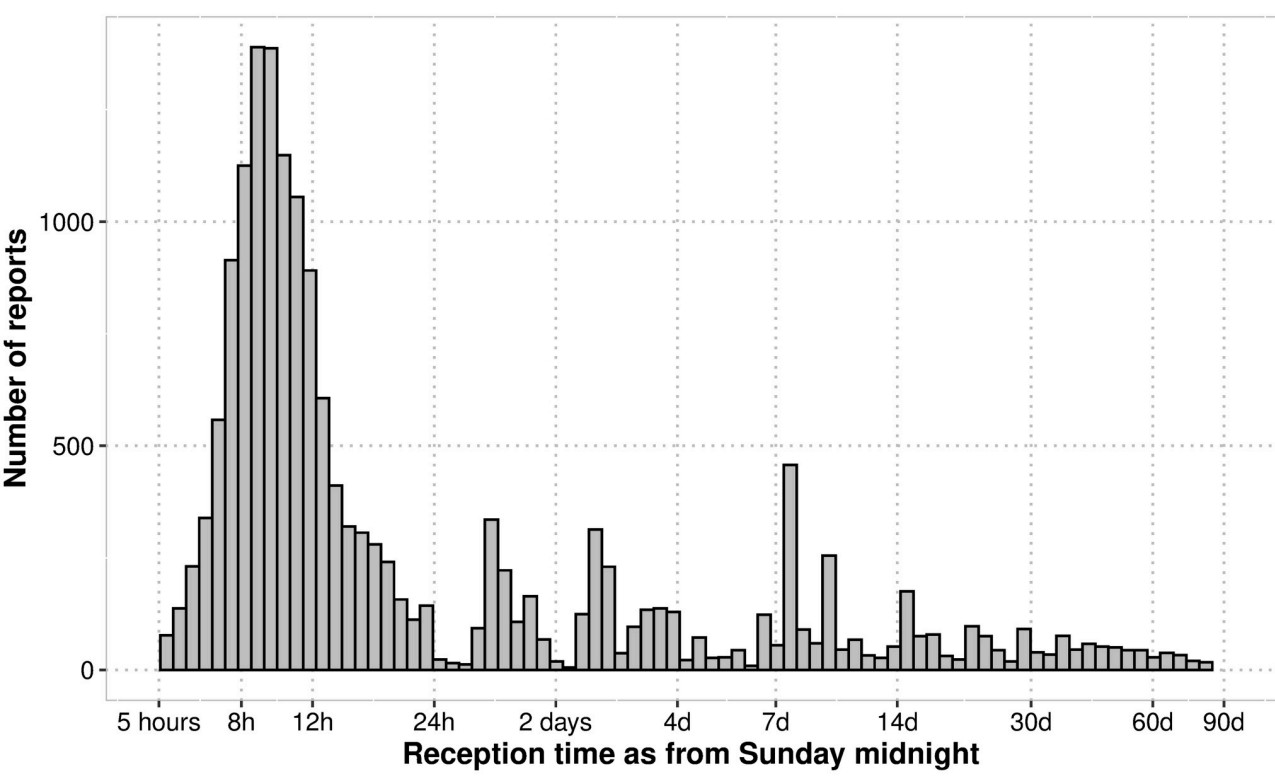

**Fig 3. Distribution of weekly reports reception time as from Sunday midnight (n = 17221 received reports).** Reception time axis: base 10 log scale.

between the phones and the central server for an estimated overall cost of 31 870 USD (cost of 0.033 USD per SMS using network 1 and 0.05 USD per SMS using network 2). Considering 15 diseases and conditions to be reported, the cost to send a weekly report and receive feedback was 1 USD using network 1 and 1.5 USD using network 2.

## Discussion

The use of Argus in Togo enabled healthcare facilities to send their weekly reports and alerts through SMS in a user-friendly, reliable and quick manner. During an almost three years period, the overall completeness of data reporting by healthcare facilities was 76%. Eighty percent of the reports of a given week were received by Tuesday evening of the following week. Automated feedback to the healthcare facilities on reached alert thresholds, automated highlights of these reached thresholds on the web platform, and automated data analyses facilitated quick identification of potential acute public health events.

Through the Argus web platform, each level could perform its respective duties. While 89% of reports where quickly reviewed by the surveillance officers at the district level, surveillance officers at the regional and central levels showed a lack of engagement with only 68% and 35% of their respective aggregated reports reviewed. Re-engagement of surveillance officers at all levels during the reinforcement phase enabled a dramatic increase in completeness and timeliness of data report and data review, especially at the central level.

With three years of operation in two regions of the country, our study was able to provide a description of the use of Argus for both an extended time-period and area [15, 16]. The lack of monitoring of the completeness and timeliness of weekly reporting by healthcare facilities in the paper-based system prevented temporal comparisons before and after the implementation

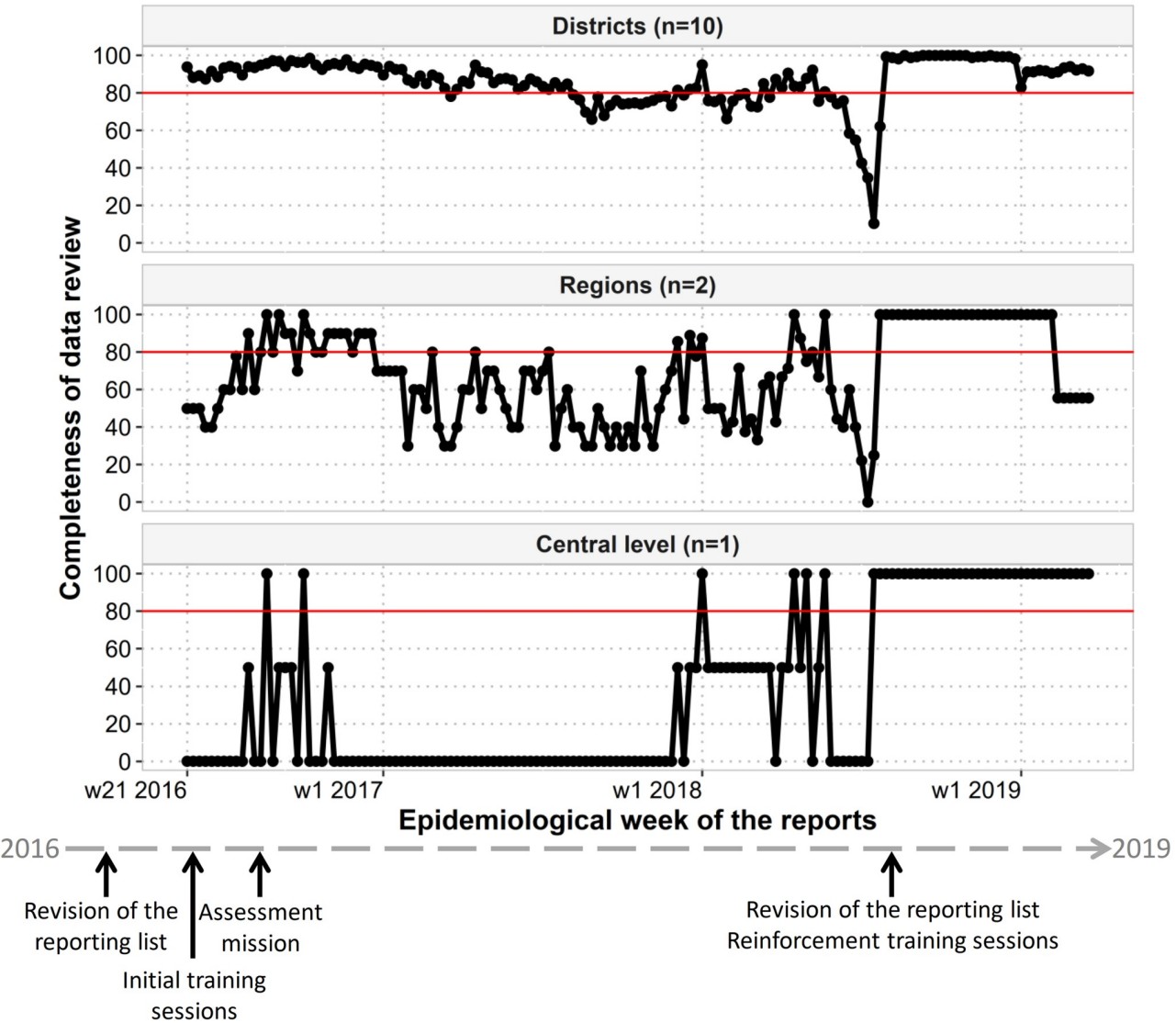

**Fig 4. Evolution of completeness of data review from week 21 2016 to week 12 2019 by level.** The drop on week 30 of 2018 was linked to the update of the system to accommodate the new list of diseases to be reported.

of Argus and geographical comparisons with regions without Argus. Indeed, only figures of completeness and timeliness of data reporting from the district level were available in the paper-based system. This capacity of Argus to monitor the number of reporting healthcare facilities provided the capacity to display the representativeness of each aggregated report.

We believe the revision by the Ministry of Health of the aggregated data to be weekly reported, with a decrease in the number of diseases and conditions from 20 to 13 and in the number of variables from 17 to 2 was crucial in the adhesion of healthcare facility staff. Due to the study design, the effect of this reduction in the completeness and timeliness of data reporting can't be independently assessed from the use of Argus, but is very likely important. The crucial role of supervision at district, regional and central level was highlighted by the relationship between completeness and timeliness of data reporting and data review. It showed that better performance could be achieved when each level took upon its duties in a timely manner.

**Table 3. Results of the evaluation questionnaires.**

| Indicator | No. of respondents | Score (from 1 worst to 5 best) | |
|---|---|---|---|
| | | Median | 10th—90th percentiles |
| Argus Android client used at healthcare facilities | | | |
| General appearance | 143 | 5.0 | [4.0; 5.0] |
| Available documentation | 142 | 5.0 | [4.0; 5.0] |
| Usefulness for data reporting | 143 | 5.0 | [4.0; 5.0] |
| Overall simplicity of use | 142 | 5.0 | [4.0; 5.0] |
| Simplicity of use for data reporting | 142 | 5.0 | [4.0; 5.0] |
| Argus web platform used by surveillance officers | | | |
| General appearance | 14 | 4.5 | [3.3; 5.0] |
| Available documentation | 16 | 5.0 | [4.0; 5.0] |
| Usefulness for data review | 15 | 5.0 | [4.0; 5.0] |
| Usefulness for data analysis | 15 | 4.0 | [2.4; 5.0] |
| Overall simplicity of use | 16 | 5.0 | [2.5; 5.0] |
| Simplicity of use for data review | 15 | 5.0 | [3.4; 5.0] |
| Simplicity of use to monitor completeness and timeliness of data reporting | 16 | 5.0 | [3.0; 5.0] |

When a drop in the completeness of data review happened at the central and regional level, a drop was also identified in the completeness of data reporting. However, completeness of data reporting remained high in the mainly rural and hard-to-reach region of Savanes. This highlights the added value of data transmission by SMS in remote areas [4].

The main challenge faced during the study was the lack of involvement and ownership of the central level on a day to day basis, as already described in similar settings [5, 7]. Reinforcement training and designation of a responsible focal point at the central level at the onset of the reinforcement phase lead to increased performance at all levels.

Another common challenge was the lack of available IT competencies at the Ministry of Health [5]. The Argus web platform was offline for three extended periods due to configuration problems with the Internet service provider. These offline periods could have been reduced to a few hours had qualified IT staff at the Ministry of Health been available and responsible for the system. While some authors described network coverage and internet availability at district level as challenges [7], these did not significantly affect the use of Argus in Togo.

With an implementation cost of 23 760 USD for 148 facilities Argus was an affordable solution. Communication costs could easily be reduced through specific agreements with mobile service providers and through the decrease in the number of SMS used to report (currently one SMS is used for each disease). While we couldn't compare the cost of using Argus with the cost of the previous paper-based reporting system, several authors reported fewer costs in a system using an IT tool mainly due to the removal of paper transportation and data entry costs [4, 13, 17, 18].

We believe providing a tool that complies with existing IDSR procedures in the country, especially its workflow trough the different levels of the surveillance system, is a prerequisite to its adoption and sustainability. Monitoring the work of surveillance officers at each level and displaying the results in a shared platform was an effective manner to make every actor accountable. Simplicity of use and limitation to features needed for each actor's duties were at the core of Argus development. This simplicity by design may explain the limited time needed to fill a report (4 min) and the high accuracy of the reported data (99.7% consistency with paper forms).

A common criticism in the field of mHealth is the lack of interoperable and open sourced systems that are flexible enough to interface with existing infrastructure (e.g. existing health information management system) and to adapt to new requirements (e.g. new list of diseases to be reported) [5, 9, 19, 20]. Through its modular architecture and use of XML files to share information Argus can easily be used with other existing tools [21]. At the Ministry of Health, both the units in charge of the IDSR and of the Health Information Management System were involved in the pilot of Argus. The unit in charge of Health Information Management System was also piloting DHIS2 as a data warehouse and monthly data reporting tool. One option considered by the authorities was to automatize a weekly data export from the Argus instance to the DHIS2 instance.

Similarly, unlike elsewhere reported [5], Argus was conceived from the beginning to be highly adaptable to the specific procedures of each country. Its administrators can easily set up the number of levels of the system, the list of diseases and variables to be reported, or activate and deactivate reporting sites over time. Its free and open-source license enables Argus to be used free of charge and to be modified to best suit the needs of each country [18].

Argus has proved itself an affordable and effective solution for public health surveillance in the two regions of Togo evaluated in this study. Extension of Argus to the whole country is currently under consideration by the Togolese health authorities. Such extension should only be performed ensuring health authorities at each level of the system are fully engaged in the daily operation of the system and staff with IT skills is available as needed. When considering the use of telecommunication and IT tools to improve public health surveillance, countries must also consider other crucial factors such as the integration of data collection and reporting between parallel systems and the limitation in the amount of collected data. Indeed, while the burden put by health information systems on healthcare facility staff is often overwhelming [22–24], minimum data (i.e. weekly number of cases, or alert of an unexpected or serious event) is enough to trigger an investigation.

## Supporting information

**S1 Questionnaires. Evaluation questionnaires.**
(PDF)

**S1 Dataset.**
(ZIP)

## Acknowledgments

We thank Lisa Stevens for reviewing the paper.

## Author Contributions

**Conceptualization:** José Guerra, Kokou Mawule Davi, Pierre Nabeth.

**Data curation:** José Guerra.

**Formal analysis:** José Guerra.

**Funding acquisition:** Pierre Nabeth.

**Investigation:** José Guerra, Kokou Mawule Davi, Florentina Chipuila Rafael, Hamadi Assane, Tsidi Agbeko Tamekloe, Aklagba Kuawo Kuassi, Farihétou Ouro-kavalah.

**Methodology:** José Guerra.

**Project administration:** José Guerra.

**Software:** José Guerra.

**Validation:** José Guerra.

**Visualization:** José Guerra.

**Writing – original draft:** José Guerra.

**Writing – review & editing:** José Guerra, Kokou Mawule Davi, Florentina Chipuila Rafael, Hamadi Assane, Lucile Imboua, Fatoumata Binta Tidiane Diallo, Tsidi Agbeko Tamekloe, Aklagba Kuawo Kuassi, Farihétou Ouro-kavalah, Ganiou Tchaniley, Nassirou Ouro-Nile, Pierre Nabeth.

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
