## [Decision Letter · Decision Letter 0]

13 Sep 2020

PONE-D-20-01670

Argus in Togo: an SMS and web-based application to support weekly public health surveillance, results from 2016 to 2019.

PLOS ONE

Dear Dr.Guerra

Thank you for submitting your manuscript to PLOS ONE. After careful consideration, we feel that it has merit but does not fully meet PLOS ONE’s publication criteria as it currently stands. Therefore, we invite you to submit a revised version of the manuscript that addresses the points raised during the review process.

We look forward to receiving your revised manuscript.

Kind regards,

Luigi Lavorgna

Academic Editor

PLOS ONE

Journal Requirements:

2. Our internal editors have looked over your manuscript and determined that it is within the scope of our Digital Health Technology Call for Papers. This collection of papers is headed by a team of Guest Editors for PLOS ONE: Eun Kyoung Choe (University of Maryland, College Park), Chelsea Dobbins (University of Queensland), Sunghoon Ivan Lee (University of Massachusetts, Amherst), and Claudia Pagliari (University of Edinburgh).

The Collection will encompass a diverse range of research articles on digital health technologies ranging from technology design to patient care and health systems management.  Additional information can be found on our announcement page: https://collections.plos.org/s/digital-health-tech.

If you would like your manuscript to be considered for this collection, please let us know in your cover letter and we will ensure that your paper is treated as if you were responding to this call. If you would prefer to remove your manuscript from collection consideration, please specify this in the cover letter.

3. In ethics statement in the manuscript and in the online submission form, please provide additional information about the database used in your retrospective study. Specifically, please ensure that you have discussed whether all data were fully anonymized before you accessed them and/or whether the IRB or ethics committee waived the requirement for informed consent. If patients provided informed written consent to have their data used in research, please include this information.

Reviewers' comments:

Reviewer's Responses to Questions

**Comments to the Author**

1. Is the manuscript technically sound, and do the data support the conclusions?

Reviewer #1: Partly

Reviewer #2: Yes

2. Has the statistical analysis been performed appropriately and rigorously? 

Reviewer #1: N/A

Reviewer #2: N/A

3. Have the authors made all data underlying the findings in their manuscript fully available?

Reviewer #1: No

Reviewer #2: Yes

4. Is the manuscript presented in an intelligible fashion and written in standard English?

Reviewer #1: Yes

Reviewer #2: Yes

5. Review Comments to the Author

Reviewer #1: The article presents data on the use of an SMS-based surveillance tool from 2016 thru 2019 in two regions within Togo. The goal of the system is to improve reporting of notifiable diseases to the district, regional, and central Ministry of Health in support of IDSR. The SMS tool supports the use of ISDR at all levels of public health. The tool is open source, works with an Android platform, analyzes data using R, and utilizes XML files.

The evaluation of the system is descriptive. The examination did not directly compare paper-based reporting to electronic-supported reporting. Furthermore, the examination did not compare the initial phase with the reinforcement phase. The paper largely presents a collection of data gathered by the team who worked on the implementation. The lack of rigorous methods and limited ability to generalize findings detracts from an interesting report of a novel tool developed for ISDR. While interesting and potentially useful, there are a number of items that need to be addressed before the paper can be further considered for publication.

MAJOR ITEMS

1. A major weakness of the paper is the lack of a clear study design. The paper was submitted as a Research Article. Yet the paper does not appear to be a research study. The paper should follow the STARE-HI guidelines for reporting evaluations of health informatics applications (https://www.imia-medinfo.org/new2/Stare-HI_as_published.pdf). Many aspects of STARE-HI are present in the paper. However, methodological details on study design and evaluation criteria are missing. The paper needs to make it clear in the ABSTRACT and METHODS sections that this is a descriptive study. It further needs to characterize the approach to evaluation. Why was the Evaluation mission so short in duration, even though the paper presents data on a longer time period?

With respect to Table 1 in the METHODS, a variety of time periods and measures were used. The table is a handy reference. Yet it is not clear why so many components were used. A clearer vision for the overall evaluation strategy would be helpful. Did the team use a logic model? Were there primary outcomes of interest? Which metrics were most important to the Ministry? How about the health care facilities? Districts? Were the various stakeholders consulted about the evaluation? Who organized and directed the evaluation?

2. With respect to development of the Argus tool, who commissioned the tool? WHO? Were the stakeholders like MOH and districts and health care facilities consulted about the development of the tool? Was there any kind of prototype development or usability testing done before implementing in Togo? Were best practices for e-health development utilized? Is the tool part of Togo’s e-Health Strategy as a country? It is not clear why the tool was tested in Togo first rather than another nation.

3. The paper needs a Limitations section where the authors make it clear that improvements in reporting might have been confounded by the fact that the Ministry constrained weekly reporting before the rollout of the tool. Dropping the number of diseases to be reported and reducing the fields to be reported likely facilitated better reporting even without the Argus tool. What variables were dropped from the old paper forms? Why did the ministry constrain reporting for Argus deployment? Did reporting change in the other regions? While a comparison with other regions might not be possible due to data limitations, the fact that close supervision at district, regional, and central levels was associated with better reporting suggests that the Argus tool might have little to do with the outcome. Instead if countries spent more time on these fundamental aspects of IDSR then we might have a more resilient surveillance infrastructure. This should be discussed.

4. The number of cases needed for an alert to occur was very small (primarily 1 case). Did an abundance of alerts (N=936 which is approximately 6 per week) contribute to the lackluster use of the tool at the Regional and Central levels? Syndromic surveillance tools have long suffered from “alert fatigue” in the same way as clinical decision support systems. What effect, if any, might this have had on the adoption and use (and enthusiasm) of the Argus tool outside of the health care facilities? This kind of discussion needs to be added to the DISCUSSION section.

5. The paper makes a point that Argus is open source and designed with interoperability in mind. Yet the paper does not discuss how the tool does or does not integrate with DHIS2, a system used in many countries to support IDSR. Does this tool interface with DHIS2? Are there plans to support use of Argus in conjunction with DHIS2 if some regions in a decentralized country want to use DHIS and others wish to use Argus?

MODERATE ITEMS

6. The drop in Week 30 2018 is linked to a system update that included new diseases. Did this correspond with additional training for facilities and ministry staff members? This is not discussed in the DISCUSSION section. It seemed to be a major driver of utilization.

7. Some of the detailed data presented in paragraphs in the RESULTS section could become tables that would more succinctly present the results, especially with respect to completeness and timeliness overall and stratified by district, region, and central levels. Furthermore, the differences before and after the reinforcement phase in a table aligned with the initial phase would strengthen comprehension and comparison of these data.

MINOR COMMENTS / SUGGESTIONS

8. The data on phone and computer ownership is not useful as presented. These results are not discussed, nor do these data seem to relate to completeness or timeliness. Consider removing these data.

9. It would be VERY helpful if the “Evaluation mission” as well as the reinforcement phase were highlighted in Figures 2, 3, and 4. Identifying when the re-training occurred would be helpful to see if this was associated with an uptick in completeness.

10. The images are quite grainy in Figure 1. As a result it is difficult to read the words in the images. Higher quality images should be provided for publication.

Reviewer #2: I found this paper and the described approach extremely interesting and complete. This kind of mHealth is even more actual in this pandemic phase, during which less progressed countries might have been poorly estimated due to underreporting of events.

What the authors clearly state is the limitation due to lack of engagement. Re-engagement of surveillance officers

enabled an increase in completeness of data report . It would be interesting to explain more extensively how re-engagement was performed and which strategies might be implemented to overcome lack of engagement.

6. PLOS authors have the option to publish the peer review history of their article (what does this mean?). If published, this will include your full peer review and any attached files.

Reviewer #1: **Yes: **Brian E. Dixon, PhD

Reviewer #2: No

---

## [Author Response · Author response to Decision Letter 0]

12 Nov 2020

Please find in the attached file labeled "Response to Reviewers" detailed response to each point raised by the reviewers.

---

## [Editor Report · Decision Letter 1]

17 Nov 2020

Case study of Argus in Togo: an SMS and web-based application to support public health surveillance, results from 2016 to 2019.

PONE-D-20-01670R1

We’re pleased to inform you that your manuscript has been judged scientifically suitable for publication and will be formally accepted for publication once it meets all outstanding technical requirements.

Kind regards,

Luigi Lavorgna

Academic Editor

PLOS ONE
---

## [Editor Report · Acceptance letter]

19 Nov 2020

PONE-D-20-01670R1 

Case study of Argus in Togo: an SMS and web-based application to support public health surveillance, results from 2016 to 2019. 

Dear Dr. Guerra:

I'm pleased to inform you that your manuscript has been deemed suitable for publication in PLOS ONE. Congratulations! Your manuscript is now with our production department. 

Kind regards, 

on behalf of

Dr. Luigi Lavorgna 

Academic Editor

PLOS ONE